# Near-Infrared Spectroscopy (NIRS) as a Tool for Classification of Pre-Sliced Iberian *Salchichón,* Modified Atmosphere Packaged (MAP) According to the Official Commercial Categories of Raw Meat

**DOI:** 10.3390/foods10081865

**Published:** 2021-08-12

**Authors:** Alberto Ortiz, Lucía León, Rebeca Contador, David Tejerina

**Affiliations:** Meat Quality Area, Center of Scientific and Technological Research of Extremadura (CICYTEX-La Orden), Junta de Extremadura, Ctra, A-V, Km372, 06187 Guadajira, Spain; alberto.ortiz@juntaex.es (A.O.); lucia.leon@juntaex.es (L.L.); rebecontro@gmail.com (R.C.)

**Keywords:** PLS-DA, LDA, SIMCA, authenticity, genetic background, feeding regime, packaging

## Abstract

This study evaluates near-infrared spectroscopy (NIRS) feasibility in combination with various pre-treatments and chemometric approaches for pre-sliced Iberian *salchichón* under modified atmosphere (MAP) classification according to the official commercial category (defined by the combination of genotype and feeding regime) of the raw material used for its manufacturing (*Black* and *Red* purebred Iberian and Iberian × Duroc crossed (50%) pigs, respectively, reared outdoors in a *Montanera* system and *White* Iberian × Duroc crossed (50%) pigs with feed based on commercial fodder) without opening the package. In parallel, NIRS feasibility in combination with partial least squares regression (PLSR) to predict main quality traits was assessed. The best-fitting models developed by means of partial least squares discriminant analysis (PLS-DA) and linear discriminant analysis (LDA) yielded high discriminant ability and thus offered a tool to support the assignment of pre-sliced MAP Iberian *salchichón* according to the commercial category of the raw material. In addition, good predictive ability for C18:3 n-3 was obtained, which may help to support quality control.

## 1. Introduction

Iberian dry-cured products are valuable meat products that have great importance in the Spanish diet [1] as well as in the European market [2] because of their sensorial and nutritional qualities. However, dry-cured products’ quality traits are dependent on intrinsic characteristics of the raw material used for its manufacturing, which in turn are influenced by the production factors to which the animals are subjected, such as genetic background (Iberian purebred, Iberian crossed with Duroc breed), rearing systems (outdoor or indoor rearing) and feed provided, especially during the finishing period (based on natural resources, i.e., mainly acorns and grass—*Montanera* system—vs. commercial fodder) [3,4].

The different qualities degrees resulting from the combination of genotype and feeding regime aforementioned are regulated by the current legislative framework—the Spanish Iberian Quality Standard (IQS) [5]. The IQS establishes four official quality degrees, which are represented by different commercial labels: “*Black*” (purebred Iberian pigs finished in *Montanera*—a typical free-range system in the southwest of Iberian Peninsula with feeding based on acorns and grass), “*Red*” (at least 50% Iberian-breed pigs finished in *Montanera*), “Green” (at least 50% Iberian-breed pigs in an outdoor system and fed based on commercial fodder and/or acorns and grass) and “*White*” (at least 50% Iberian-breed pigs indoor-reared and commercially fed). However, these quality standards only apply to fresh meat and dry-cured shoulder, ham and loin, excluding other Iberian meat products such as dry-cured sausages [5].

The inclusion of dry-cured sausages in the various quality standards stated by the IQS would provide information to consumers regarding the quality dimension of the product. Additionally, authentication of the quality category of these products should be supported by fast, sensitive, non-destructive and accurate control tools, such as near-infrared reflectance spectroscopy (NIRS). Indeed, proof has been shown of the NIRS’ capability to discriminate Iberian pig carcasses, subcutaneous fat, fresh meat [6] and even dry-cured loin [7] according to various quality categories stated by the IQS [5]. However, to our knowledge, the feasibility of NIRS as a qualitative approach for the classification of Iberian dry-cured sausages according to the quality category of the raw material used for its manufacturing has not been addressed yet.

On the other hand, products that are pre-sliced and packaged under vacuum or modified atmosphere (MAP) are preferred compared to the whole pieces since these are more convenient with the current purchasing habits and trends of consumers [8]. Specifically, the MAP format is the one most chosen by consumers since its appearance is more similar to that of a just sliced product, and it also avoids the adherence between slices that is typical of vacuum packaging [9]. However, in this type of selling format, stricter control of traceability is necessary in order to avoid fraudulent practices since they do not have the label present on the whole piece as a guarantee of the commercial category. 

Additionally, NIRS technology could also be used for the prediction of the main quality traits of Iberian dry-cured sausages to allow a rapid quality assessment. Indeed, NIRS has previously been used for rapid determination of the fatty acids in Iberian dry-cured sausages such as *salchichón* and *chorizo* [10] as well as for tocopherols, fatty acid content and lipid oxidation index in vacuum packaged dry-cured ham [11], reporting good predictive ability. NIRS technology may also be suitable for pre-sliced packaged dry-cured sausages under MAP format, in which the quality of the products might be altered to a greater extent than in the vacuum format [8], thus requiring more frequent quality control.

Thus, the implementation of a comprehensive system to ensure the traceability and quality control of new-selling formats of pre-sliced dry-cured sausages through NIRS technology would be of great relevance for the manufacturing industries of the Iberian sector as an anti-fraud tool and in order to make it possible to guarantee the quality of each package.

Thus, the purpose of this research was to determine the potential of NIRS technology to classify the pre-sliced MAP Iberian *salchichón* according to the official commercial category of the raw material used for its manufacturing, using various chemometric approaches. In parallel, its accuracy in the prediction of main meat quality traits was evaluated.

## 2. Materials and Methods

### 2.1. Iberian Salchichón Samples and Packaging

A total of 185 100 g packages under MAP of Iberian dry-cured *salchichón* were manufactured from meat and fat from three official commercial categories; *Black* (*n* = 59), *Red* (*n* = 63) and *White* (*n* = 63) [5] were used in the current study. Samples were randomly purchased from an Iberian industrial meat processing plant (Sierra de Barbellido, Salvaleón, Badajoz, Spain).

The production conditions to which the animals were subjected according to commercial category were as defined in IQS [5] and are summarised in Appendix A.

Iberian *salchichón* sausages were elaborated from an initial mixture composed of 96% of meat mass (65% of lean and 35% of backfat—from *Black*, *Red* and *White* commercial categories) and 2 g/100 g of NaCl, 0.4 g/100 g of commercial mixture of spices (ground dry garlic, *black* pepper, ground nutmeg) and 1.6 g/100 g of preservers (dextrose, lactose) and stabiliser (E-250, E-252, E-331) additives specially prepared for this type of meat product. After mixing, the batters were stuffed into 6–7 cm diameter natural casing, fermented and dried in a drying chamber according to standards practices of the Iberian manufacturing industry for a total of 120, 110 and 100 days, weighing 1.1 ± 0.2, 1.2 ± 0.1 and 1.4 ± 0.3 kg per piece from *Black*, *Red* and *White* category, respectively. Subsequently, Iberian *salchichón* sausages were sliced (2 mm thickness) using an industrial slicer in a clean-room slicing plant and homogenously distributed in 100 g packages under modified atmosphere (70% N_2_ and 30% CO_2_) packaging with the help of an industrial packaging machine (Ulma^®^ SMART 300). The material of the packaging used was polystyrene (150 mm thick) with an oxygen permeability of 3.2 cm^3^ O_2_/m^2^/24 h/atm at 4 °C and sealed with 70 mm thick polyethene film (VIDUCA, Alicante, Spain) with an oxygen permeability of 1 cm^3^/m^2^/24 h (4 °C; 50% RH), 5.5 cm^3^/m^2^/24 h (4 °C; 50% RH) to CO_2_ and 2.2 g/m^2^/24 h (4 °C; 90% RH) to H_2_O.

Each of the packages was stored in refrigeration chamber (4 °C) in darkness and sampled at: T0, T4, T8 and T12 (initial, 4, 8 and 12 months of storage, with 58, 54, 56 and 17 samples, respectively). Samples from T12 collapsed, preventing an adequate NIRS measurement from being performed, so the total number of them was lower in the assay. The distribution of samples in the calibration and validation sets is shown in Table 1.

### 2.2. NIRS Spectra Acquisition and Spectral Pre-Treatment

Spectral data were taken as reflectance with a LabSpec 2500 (ASD Inc., Boulder, CO, USA) NIRS spectrometer equipped with an ASD fibre-optic contact Probe^®^ (21 mm window diameter). NIR spectrometer was calibrated using a spectralon tile covered with the same material with which the product was packaged, as the purpose of the current study was to obtain predictive NIRS models for unopened packages. One spectrum (average of 40 scans over the range of 1000 to 2500 nm) per sample was acquired by direct contact sensor (ASD contact probe^®^) sample (package surface) (Appendix A). The spectrum was taken by making a zigzag sweep over the entire surface of the package to extend the sampling surface area, minimising the sampling error. Instrument monitoring and initial spectral management were performed with Indico TM Pro software package (Analytical Spectral Device-ASD Inc., Boulder, CO, USA). For further processing and chemometric analysis, data were exported into Unscrambler X vs. 10.5 (CAMO^®^, Trondheim, Norway) software.

The full spectral data were split into calibration and external validation sample sets (75% and 25% of the total of samples, respectively) by manual and random selection in order to ensure the representation in each subset of samples from all commercial categories (*Black*, *Red*, *White*) and sampling storage times (T0, T4, T8, T12). It allowed the maximisation of the variability of calibration and external validation sample sets (Table 1).

During the performance evaluation of models, outliers identified were removed. Outlier detection was carried out by detecting anomalous samples in the spectral plot by principal component analysis (PCA) since it is the most-used tool for data exploration and pattern recognition. The criteria for deleting outliers were (1) samples with residuals higher than 2; (2) samples with leverage (H) higher than 3 times the average leverage [12]:H = 1/(*n* + (number of principal components)/*n*)(1)
where “*n*” is the number of samples. 

Various pre-treatments and combinations of them were used to optimise the accuracy of calibration models: none or absorbance (Log 1/R, being R the reflectance), Standard Normal Variate (SNV), Detrending (DT) [13] and two different derivative math treatments. The SNV centres each spectrum and then scales it by its own standard deviation, correcting shifts on the log (1/R) axis. DT was performed using a second-order polynomial in a regression analysis, in which spectral values work as dependent variables whilst the independent ones are given by the wavelengths, correcting effects due to baseline curvature. With refer to derivatives, first and second-order Savitzky–Golay derivatives were performed [14]. Specifically, for the first-order derivative, 4 smoothing left- and right-side points (symmetric Kernel) and first polynomial order (1,4,4,1) were established, and for the second one, 5 smoothing left and right-side points and second polynomial order (2,5,5,2) were set.

All pre-treatments and combinations of them were applied to the spectral range between 1000 and 1800 nm since spectra displayed low signal/noise ratio from 1800 to 2500 nm (Appendix A), so wavelengths above 1800 nm were discarded.

### 2.3. NIR Qualitative Analysis

Within the aim to classify pre-sliced MAP Iberian *salchichón* according to various official commercial categories of raw material used for its manufacture (*Black*, *Red* and *White*), several qualitative approaches were evaluated: partial least squares discriminant analysis (PLS-DA) and linear discriminant analysis (LDA) as discriminant classification techniques, and soft independent modelling of class analogies (SIMCA) as class-modelling technique (Unscrambler X vs. 10.5 software (CAMO^®^, Trondheim, Norway)).

Discriminant classification studies different mathematical applications that can assign a sample to a predefined class, whilst class-modelling models look for one that confirms whether or not the sample belongs to a given class.

#### 2.3.1. PLS-DA 

PLS-DA is a supervised classification method, commonly used as the reference method in adulteration, authenticity and traceability studies of foodstuffs. Specifically, in the current study, PLS-DA models were performed to classify pre-sliced MAP Iberian *salchichón* according to various official commercial categories of raw material (*Black*, *Red* and *White*), throughout correlating spectral variations (X) with these above-mentioned defined classes (Y). For this approach, Y variables act as “dummy” variables [15] since they are not continuous, as they are in quantitative analysis. Thus, a value of 1 was assigned to samples belonging to the class, whilst 0 was used when not, assuming that this regression method can be used for qualitative purposes by calculating a calibration model that links the predictor matrix and this dummy response matrix. Goodness of models was assessed based on the highest value of the determination coefficient (1-VR) and the lowest root mean square error after cross-validation (RMSECV) (performed by leave-one-out method), as well as sensitivity (SE) and specificity (SP) in both calibration and external validation sample sets [16]:-SE refers to the percentage of samples of a given class that the model correctly recognises as belonging to that class:
SE = TP/(TP + FN)(2)
-SP refers to samples that do not belong to a given class and are correctly rejected by the model:
SP = TN/(TN + FP)(3)
where TP = true positives, FN = false negatives, TN = true negatives and FP = false positives.

Results derived from all pre-treatments evaluated are compiled in Appendix A, whilst results of the best-fitting models are summarised in Table 2.

#### 2.3.2. SIMCA 

SIMCA is a class-modelling approach in which each class (*Black*, *Red* and *White*) is modelled independently throughout a PCA model. The dimension of each individual PCA model is given by the number of principal components (PCs). Subsequently, sample may be classified into one, various or none of the above-mentioned classes, depending on the distance of the sample to the centre of the model (leverage) and the distance of the sample to the model defined by the PCs (S-distances). In the case that a sample belongs to two or more classes, it will be assigned to the class with the lowest values of both leverage and S distances.

Classification ability of the models was expressed in terms of SE and SP [16]. Results derived from all pre-treatments evaluated are compiled in Appendix A, whilst results of the best-fitting models are summarised in Table 3.

#### 2.3.3. LDA

LDA analysis is based on the description of the data by probability density distributions, under the assumption that they are multivariate normal and have equal dispersion and correlation between variables within all classes (*Black*, *Red* and *White* in the current study). The major limitation of LDA is the need to have more rows than columns in the data matrix, as in our case. Therefore, to overcome this hurdle, LDA could be applied after a preliminary dimension reduction by PCA since it is a well-known data compression method commonly used for dimension reduction of spectral variables [17]. Such approach would allow to execute LDA using scores on the significant PCs as new descriptor variables [15]. The main aim of LDA is to develop a procedure for predicting class membership for new samples, finding vectors that lead to maximum separation among classes, which are achieved by maximising the ratio of the between-class variance to the within-class variance. Mahalanobis method was used as a measure of class distance, and the number of PCs was optimal suggested by the PCA model. The evaluation of the LDA models was based on the above-mentioned parameters SE and SP [16]. Results derived from all pre-treatments evaluated are compiled in Appendix A, whilst results of the best-fitting models are summarised in Table 4.

### 2.4. Quantitative NIR Analysis

For the quantitative prediction of the quality traits, independent partial least squares PLSR models were developed for dry matter (DM), chloride content (NaCl), tocopherols (α- and γ-) content, lipid (mg MDA/kg) and protein oxidation (nmol Carbonyls/mg) and C16:0, C18:0, C18:1 n-9, C18:2 n-6 and C18:3 n-3 fatty acids. The performance of the models was evaluated using leave-one-out full internal cross-validation. Spectral data subjected to PLS produce a new and smaller set of variables called “latent variables” (LV), for the which the optimal number was reflected as the number of LVs after which the standard error of cross-validation (SECV) no longer decreased substantially. Goodness of predictive models was assessed based on the highest value of 1-VR and the lowest RMSECV and LV. For each parameter, the external validation was performed, reporting the coefficient of determination of external validation (R^2^v) and root mean square error of external validation (RMSEV). Descriptive statistics of quality traits evaluated are presented in Table 5, whilst results of the best-fitting quantitative models are summarised in Table 6. Results derived from all pre-treatments are compiled in Appendix A. Furthermore, for the best-fitting calibration models (Table 6), the ratio between the standard deviation (SD) of the set of samples and the SECV (SD/SECV, known as the residual prediction deviation (RPD) index) [18], as well as the relationship between the interval of the composition of the reference data for the collective calibration (Y_max_ − Y_min_) and the SECV, known as the range error ratio (RER) index, were calculated, since they are considered statistics indicators with the greatest weight in the precision of an NIRS model [19].

### 2.5. Reference Analysis

Chemical determinations of the above-mentioned quality traits considered for quantitative NIR analysis were carried out as follows. Thus, DM was assayed following the AOAC method [20] and NaCl content using the Volhard method [21]. In both, the results were expressed as g/100 g of sample.

α- and γ-tocopherol content were measured using the method proposed by Liu, Scheller and Schaeffer [22]. Extracting and HPLC conditions are widely described by García-Torres, Contador, Ortiz, Ramírez, López-Parra and Tejerina [4]. Results were expressed as µg/g of α- or γ-tocopherol/g sample.

Lipid oxidation was evaluated by the 2-thiobarbituric acid (TBA) method [23]. TBA values were calculated from the standard (1,1,1,3-tetraethoxypropane, TEP) curve and expressed as mg malondialdehyde (MDA)/Kg Iberian *salchichón*. Protein oxidation was assessed by measurement of carbonyl groups formed during incubation with 2,4-dinitrophenylhydrazine (DNPH) in 2N HCl [24]. Protein concentration was calculated by spectrophotometry at 280 nm using bovine serum albumin (BSA) as standard and expressed as nmol carbonyls/mg protein.

The fatty acid profile was determined from the fat extracted [25]. The chromatographic conditions are described in detail by García-Torres et al. [4]. Results were expressed as g/100 g of fatty acid methyl esters (FAMEs).

The main descriptive statistics of all these quality traits are shown in Table 5.

## 3. Results

### 3.1. Exploration of the Spectral Data

Appendix A shows the raw spectra (reflectance) data of unopened pre-sliced MAP Iberian *chorizo* between 1000 and 2500 nm of calibration and external validation sets grouped by the commercial category of the raw material (*Black*, *Red* and *White*) used for its manufacture.

It can be noted that above 1800 nm, there is scarce useful spectral information, where the most useful data are available between the 1000 and 1800 nm, given that this area displayed a high signal/noise ratio. High noise was observed above 2300 nm. The patterns of spectra were similar for the various categories (*Black, Red* and *White*). Nevertheless, slight differences can be noted in intensity absorbance due to the commercial category of raw material in some area’s region around 1090, 1280 and 1640 nm. Thus, higher absorbance intensity was observed for Iberian *salchichón* samples manufactured from *White* category than those elaborated from *Black* and *Red* ones, the latter two overlapping along the spectral range (Appendix A).

### 3.2. NIRS Qualitative Predictive Models

The results of the best-fitting prediction models developed by PLS-DA, SIMCA and LDA approaches to classify pre-sliced MAP Iberian *salchichón* according to the commercial category of raw material (*Black*, *Red* and *White*) and their respective validations with an external validation sample set are summarised in Table 2, Table 3 and Table 4. Additionally, the results derived from the use of several pre-treatments, and a combination of these in each qualitative approach are presented in Appendix A.

Regarding PLS-DA, the best-fitting discriminant model was obtained after applying SNV-DE in combination with SG 1,4,4,1 (Table 2). The best 1-VR and the lowest RMSECV were attained by the *White* model (0.782 and 0.226, respectively), while the other 1-VRs did not drop below 0.59. When models were validated, SE decreased above 30% compared to the values observed in calibration, while the decline in SP was less abrupt, remaining at good levels—above 70%—for both *Red* and *White* models. However, the *Black* category yielded 45.16% of SP after external validation. When the data pattern was explored by means of the Principal Component Analysis (PCA) chemometric tool (Appendix A), it may be graphically observed how the above-mentioned pre-treatment allows separating the samples according to the different classes (*Black*, *Red*, *White*) (Appendix A). More precisely, the samples with the *Red* category were mainly placed in the negative of principal component (PC) 1, while those belonging to the *Black* category had positive scores in PC 2. The samples of the *White* category were distributed into positive PC1 scores and negative PC2 scores, thus suggesting that the classification of pre-sliced MAP Iberian *salchichón* samples according to the commercial category of raw material used may be possible after pre-treatment SNV-DE SG 1,4,4,1.

In relation to spectral features, Appendix A shows the graphical representation of regression coefficients (B) of wavelengths of Iberian *salchichón* spectral data from PLS-DA analysis after SNV-DE SG 1,4,4,1 (Absorbance (log1/R)) at 1000–1800 nm. It can be observed that the wavelengths with the highest weight (regression coefficients) for the best-fitting PLS-DA models were mainly around 1000 nm, and the range comprised between 1040 and 1100 nm and between 1200 and 1300 nm.

Excellent results (around 100%) were observed for the SE by models developed by means of SIMCA with respect to those obtained by PLS-DA. In contrast, SP results were much inferior (Table 3). More in detail, the best discriminant was obtained in absorbance (log 1/R). Calibration models showed perfect SE after external calibration for all categories. SP values were low, as expected, given the low values of SP previously observed in calibration but remained substantially unchanged from those observed in calibration.

The best-fitting discriminant equations obtained by means of LDA (Table 4) resulted after applying SNV in combination with DE pre-treatment. In general, excellent calibration results were obtained, showing a balanced pattern between both SE and SP statistics and among the commercial categories in calibration and after external validation. Thus, SE values ranged between 79.17% for *White* and 87.23% for *Red* categories in calibration, whilst SP values comprised between 87.91% for *Red* and 94.44% for *White*. Furthermore, the good SE and SP values were maintained after the external validation, with the exception of the SE of the *Black* category, in which a sharp drop was observed.

### 3.3. NIRS Quantitative Predictive Models

The main descriptive statistics of quality traits are shown in Table 5. The first evaluation indicates that all parameters studied yielded a wide range of values in both calibration and external validation sample sets, as well as similar mean and SD, indicating an appropriate representation of the variability in both sets. This is the first step to obtain good accuracy by NIRS prediction models.

For DM and NaCl, similar predictive ability was observed, with good 1-VR, RMSECV statistics yielded by the best-fitting models, which were obtained after the same pre-treatment (SNV-DE) (Table 6). Nevertheless, when these models were validated, both of them showed limited predictive ability given by the low values of R^2^v, RPDV and RERV.

The calibration models developed for antioxidants content displayed slightly higher values of 1-VR than those for fatty acids, with the exception of the model developed for linolenic acid (C18:3 n-3), for which the best statistics were observed (1-VR, RMSECV) (Table 6). In general terms, 1-VR values ranged from 0.612 for oleic acid (C18:1 n-9) to 0.824 for C18:3 n-3, and the RMSECV comprised between 0.146 for C18:1 n-9 and 1.623 for γ-tocopherol. The best predictive equations were observed after applying SNV-DE in combination with the first derivate (SG 1,4,4,1), with the exception of C18:1 n-9 and linoleic acid (C18:2 n-6), which were obtained after SG 1,4,4,1. After external validation, the model developed for C18:3 n-3 reported a great predictive capability since R^2^v and RMSEV yielded 0.808 and 0.142 values, respectively, and high RPD_v_ and RER_v_ statistics were also observed. Additionally, α-content and stearic (C18:0) displayed acceptable values of these aforementioned statistics. However, on the opposite, γ-tocopherol, palmitic acid (C16:0), C18:1 n-9 and C18:2 n-6 models reported quite low values for these statistics.

The best-fitting model developed for lipid oxidation index (mg MDA/kg) prediction yielded higher predictive ability than that obtained for protein oxidation index (nmol carbonyls/mg), with the values of 0.746 and 0.441 for 1-VR, and 0.213 and 0.343 for RMSECV, respectively (Table 6). Both of them were obtained after applying SNV-DE in combination with the first derivate (SG 1,4,4,1). Nevertheless, more satisfactory outcomes would be desirable in the validation of these models, especially for protein oxidation.

Appendix A shows how the important variables on the best PLS models developed for the above-mentioned parameters were broadly distributed over the entire spectral range between 1000 and 1800 nm.

## 4. Discussion

The collection of high-quality spectra is crucial for the construction of reliable discriminant models. Therefore, the spectral range used in the current study for chemometric analysis was that comprised between 1000 and 1800 nm, since above this wavelength, not enough signal reached the detector, resulting in low signal/noise areas with limited useful spectral information (Appendix A). Cáceres-Nevado, Garrido-Varo, De Pedro-Sanz, Tejerina-Barrado and Pérez-Marín [26] observed a similar problem but at a lower wavelength (around 1600 nm) using a MicroNIR^TM^ 1700 microspectrophotometer when collecting spectra in Iberian fresh loin.

Despite the similar spectral data shape observed, some minimal differences of absorbance intensity can be noted at the main absorption dominated bands, around 1090, 1280 and 1640 nm. These differences could be enough to classify pre-sliced MAP Iberian *salchichón* according to the commercial categories. The main absorption dominated bands were related to the second and third overtone region of the C-H bonds, which is the base of fatty acids and α- and γ-tocopherols [7,27]. Thus, the similarity between spectra from samples manufactured with raw material from the *Black* and *Red* categories and the differences between these and the ones manufactured with *White* raw material (Appendix A) would corroborate the differences caused by animal’s feeding reported in previous studies of dry-cured products [3,4]. Indeed, the spectral differences due to animal feeding regimes have recently been used to discriminate between premium (acorn-based) and non-premium (concentrate feed) Iberian carcasses [6,28], Iberian live animals, fresh meat and subcutaneous backfat [6]. Therefore, spectral differences, specifically those in the above-mentioned areas, might have allowed the classification of pre-sliced MAP Iberian *salchichón* according to the commercial category of the raw material [7].

The good accuracy in classification results by means of PLS-DA in calibration set was not unexpected, since previous studies have highlighted the good classificatory ability of NIRS technology in combination with PLS-DA for quality categories assignment support control in Iberian live animals, carcasses, subcutaneous fat, fresh meat (*psoas major muscle*) [6] and dry-cured loin [7]. Additionally, PLS-DA has been assessed as other control tools, such as the discrimination between fresh and frozen-thawed acorns-fed Iberian loins [26], offering a very high classificatory ability. However, SE values in all categories and SP for the *Black* category decreased considerably in the external validation set. Scientific literature dealing with qualitative models for classification into commercial categories compiled by the current IQS is scarce in meat [6,28] and meat products [7], probably because of the short time since its approval (in 2014) and its inexistence for Iberian dry-cured sausages. However, the results of the present study might provide the basis for the use of NIRS for discrimination purposes of Iberian *salchichón* by the commercial category of raw material.

As far as SIMCA models are concerned, although high SE values were obtained, such was not the case for SP. The ability to discriminate samples that do not belong to a certain category is of paramount importance in authenticity control tools, especially for the top-quality categories, *Black* and *Red* [29], as they have the highest prices in the market, being, therefore, the most exposed to fraudulent practices. Thus, more satisfactory outcomes by SIMCA would be desirable in terms of SP. Regarding the scientific literature, SE values obtained in the present study were similar to those obtained by Tejerina, Contador and Ortiz [7] for pre-sliced Iberian dry-cured loin and packaged under modified atmosphere, whilst SP values were quite lower. SIMCA has also been used in combination with NIR technology as an authentication tool to discriminate between perirenal fat in lambs according to the animal’s feeding regime [30] or for the identification of different animal species in ground meat [31], obtaining better SP values than those from the current study. 

Finally, classification ability obtained by LDA was in general good in terms of SE and SP in calibration and after external validation, in agreement with results obtained by Tejerina et al. [7] for pre-sliced Iberian dry-cured loin. The good and balanced pattern observed for both SE and SP and for the various categories by means of LDA suggests that this approach could provide better classification results than more sophisticated ones, as previous studies carried out in Iberian dry-cured loin [7] and perirenal fat of lambs [30] have reported. Consequently, the model obtained from LDA would be the most feasible for classifying the pre-sliced MAP Iberian *salchichón* according to the official commercial category of the raw material used for its manufacturing.

The goodness of quantitative predictive models was based on the greatest 1-VR as well as the minimum REMCV and the lowest number of LVs. For the former, a value between 0.66 and 0.80 indicates approximate quantitative predictions, whereas a value between 0.81 and 0.90 reveals a good prediction. Calibration models with 1-VR > 0.90 are considered to be excellent [32]. Moreover, in order to assess the practical utility of the prediction models, the RPD and RER statistics were considered in the external validation set. In relation to RPD, Prieto et al. [33] considered that a value between 2 and 2.5 makes approximate quantitative predictions possible and can be applied to meat products. As far as RER is concerned, values lower than 3 indicate small predictive capability, whereas a value between 3 and 10 and higher than 10 indicates moderate and good practical utility, respectively [19]. Thus, given that for all quality traits evaluated, the 1-VR value was comprised between 0.612 and 0.824, with the exception of protein oxidation index (nmol Carbonyls/mg protein), approximate quantitative predictions might be possible [32]. 

By comparing our results with previous studies, NaCl, C16:0 and C18:1 n-9 models yielded slightly lower 1-VR values than those reported by Tejerina, García-Torres, Cabeza de Vaca, Ortiz and Romero-Fernández [11] in pre-sliced vacuum packaged Iberian dry-cured ham, while the values obtained for tocopherols, C18:0, C18:2 n-6 and C18:3 n-3 were higher. Fernández-Cabanás, Polvillo, Rodríguez-Acuña, Botella and Horcada [10] evaluated the NIRS technology in combination with modified partial least squares regression (MPLSR) to predict the fatty acid profile of Iberian *salchichón* and *chorizo* sausages jointly. These authors obtained higher 1-VR values for all fatty acids than those observed in the current study, with the exception of C18:3 n-3. Discrepancies in predictive ability between studies could be related to sample preparation [34,35] since the spectra acquisition was carried out directly on minced sausages by Fernández-Cabanás et al. [10] compared to unopened packages in the current study. Thus, worse results would be expected.

After the external validation, a decline in all statistics above-mentioned discussed took place. Thus, in general, the RPD and RER limits required to ensure the reliability of the models were not exceeded [19,36]. Indeed, only the model developed for C18:3 n-3 proved to be able to predict when models were validated. The decline in the predictive ability after external validation was also observed in other studies dealing with the NIRS ability for fatty acid profile prediction [36] and could be explained by the requirement of a larger number of samples in calibration and external validation sample sets.

## 5. Conclusions

The results derived from the present study contribute to shedding light on the potential of NIRS technology for the official commercial categories assignment according to the current legislative framework and quality assessment of Iberian *salchichón* dry-cured sausage. The results obtained propose that NIRS technology, in combination with spectral pre-processing and LDA as a chemometric approach, could help against fraud of the individual pre-sliced MAP Iberian *salchichón* according to various official commercial categories of raw material (*Black*, *Red* and *White*) compiled by the current Spanish Iberian Quality Standard used for their manufacture. Furthermore, NIR technology might be useful as a control tool, especially for fatty acids such as C18:3 n-3. Further works are required to improve the predictive ability of other quality parameters such as tocopherols and lipid oxidation in order to estimate the shelf life of pre-sliced Iberian dry-cured sausages. 

The surrounding plastic packaging material and the composition of the atmosphere, however, are the main limitations to be taken into account since the results obtained in the present study could be influenced by these factors. Further works should evaluate the NIRS feasibility as an authenticity and quality control tool on other packaging types such as vacuum and active packaging as well as other Iberian dry-cured products.

## Figures and Tables

**Table 1 foods-10-01865-t001:** Distribution of samples in the calibration and validation sets according to commercial categories of raw material used for manufacturing Iberian *salchichón* and storage time.

	Sampling Storage Time	Commercial Category			Total
		*Black*	*Red*	*White*	
Calibration	T0	13	15	15	43
	T4	13	12	15	40
	T8	14	15	15	44
	T12	3	5	3	11
	Total	43	47	48	138
Validation	T0	5	5	5	15
	T4	5	5	4	14
	T8	4	4	4	12
	T12	2	2	2	6
	Total	16	16	15	47
Total		59	63	63	185

T0, T4, T8 and T12 = initial, 4, 8 and 12 months of storage, respectively. *Black*, *Red* and *White* = commercial categories of raw material according to the current Spanish Iberian Quality Standard used for manufacturing Iberian *salchichón*.

**Table 2 foods-10-01865-t002:** PLS-DA results of the best-fitting discrimination model of pre-sliced MAP packages of Iberian *salchichón* according to the commercial category of raw material.

Commercial Category	Pre-Treatment	LVs	Calibration					External Validation		
			*n*	1-VR	RMSECV	SE	SP	*n*	SE	SP
*Black*	SNV-DE SG 1,4,4,1	8	40	0.599	0.296	69.77	81.05	16	43.75	45.16
*Red*			42	0.653	0.279	78.72	80.22	16	43.75	70.97
*White*			46	0.782	0.226	91.67	92.22	15	46.67	78.13

*Black*, *Red* and *White* = commercial categories of raw material defined by the current Spanish Iberian Quality Standard used for manufacturing Iberian *salchichón*; SNV = Standard normal variate; DE = de-trending; SG = Savitzky–Golay derivates; LVs = latent variables; *n* = number of samples; 1-VR = coefficient of determination in cross-validation; RMSECV = root mean square error of cross validation; SE = sensitivity; SP = specificity.

**Table 3 foods-10-01865-t003:** SIMCA results of the best-fitting discrimination model of pre-sliced MAP packages of Iberian *salchichón* according to the commercial category of raw material.

Commercial Category	Pre-Treatment	PCs	Calibration			External Validation		
			*n*	SE	SP	*n*	SE	SP
*Black*	Absorbance	2	43	100.00	18.95	16	100.00	19.35
*Red*		1	47	95.74	3.30	16	87.50	3.22
*White*		2	48	100.00	23.33	15	100.00	21.88

*Black*, *Red* and *White* = commercial categories of raw material defined by the current Spanish Iberian Quality Standard used for manufacturing Iberian *salchichón*; PCs = number of principal components; *n* = number of samples; SE = sensitivity; SP = specificity.

**Table 4 foods-10-01865-t004:** LDA results of the best-fitting discrimination model of pre-sliced MAP packages of Iberian *salchichón* according to the commercial category of raw material.

Commercial Category	Pre-Treatment	Calibration			External Validation		
		*n*	SE	SP	*n*	SE	SP
*Black*	SNV-DE	43	81.40	91.58	16	75.00	80.65
*Red*		47	87.23	87.91	16	81.25	77.42
*White*		48	79.17	94.44	15	53.33	96.88

*Black*, *Red* and *White* = commercial categories of raw material defined by the current Spanish Iberian Quality Standard used for manufacturing Iberian *salchichón*; SNV = standard normal variate; DE = de-trending; *n* = number of samples; SE = sensitivity; SP = specificity.

**Table 5 foods-10-01865-t005:** Descriptive statistics of quality traits of pre-sliced MAP Iberian *salchichón*.

Parameters	Calibration					External Validation				
	*n*	Mean	Min	Max	SD	*n*	Mean	Min	Max	SD
DM (g/100 g)	138	73.13	69.09	78.04	1.36	47	73.02	69.09	78.04	1.29
NaCl (g/100 g)	138	3.58	2.91	4.46	0.32	47	3.55	2.98	4.46	0.33
alpha (µg/g)	138	10.30	4.89	17.14	3.28	47	10.37	4.89	17.44	3.35
gamma (µg/g)	138	0.85	0.18	1.93	0.24	47	0.85	0.18	1.93	0.24
C16:0 ^1^	138	24.64	20.65	26.44	0.92	47	24.59	20.65	26.44	0.97
C18:0 ^1^	138	11.59	5.36	12.69	1.24	47	11.49	5.36	12.69	1.34
C18:1 n-9 ^1^	138	51.00	48.71	58.50	1.50	47	51.07	48.71	58.50	1.60
C18:2 n-6 ^1^	138	5.74	5.05	7.11	0.36	47	5.75	5.05	7.11	0.38
C18:3 n-3 ^1^	138	0.59	0.29	1.53	0.30	47	0.62	0.29	1.74	0.33
mg MDA/kg	138	1.44	0.59	2.31	0.43	47	1.40	0.59	2.31	0.44
nmol Carbonyls/mg protein	138	3.30	2.02	4.54	0.52	47	3.32	2.03	4.54	0.52

DM = dry matter; MDA = malondialdehyde; C16:0 = palmitic acid; C18:0 = stearic acid; C18:1 n-9 = oleic acid; C18:2 n-6 = linoleic acid; C18:3 n-3 = linolenic acid; *n* = number of samples; SD = standard deviation. ^1^ Fatty acids were expressed as g/100 g fatty acid methyl esters.

**Table 6 foods-10-01865-t006:** PLSR results of the best-fitting prediction models of main quality traits of pre-sliced MAP Iberian *salchichón* samples.

Parameter	Pre-Treatment	LVs	*n*	Calibration		External Validation			
				1-VR	RMSECV	R^2^_V_	RMSEV	RPD_V_	RER_V_
DM (g/100 g)	SNV-DE	10	122	0.704	0.700	0.204	1.234	1.134	6.408
NaCl (g/100 g)	SNV-DE	10	123	0.687	0.715	NA	69.439	0.005	0.020
alpha (µg/g)	SNV-DE SG 1,4,4,1	5	117	0.730	1.522	0.601	2.029	1.600	5.998
gamma (µg/g)	SNV-DE SG 1,4,4,1	5	125	0.731	1.633	NA	9.454	0.028	0.169
C16:0 ^1^	SNV-DE SG 1,4,4,1	6	128	0.651	0.560	0.184	0.729	1.120	5.490
C18:0 ^1^	SNV-DE SG 1,4,4,1	6	128	0.728	0.597	0.554	0.673	1.514	6.437
C18:1 n-9 ^1^	SG 1,4,4,1	6	126	0.612	0.889	0.118	1.036	1.076	5.162
C18:2 n-6 ^1^	SG 1,4,4,1	6	129	0.652	0.206	0.386	0.259	1.291	6.112
C18:3 n-3 ^1^	SNV-DE SG 1,4,4,1	4	134	0.824	0.146	0.808	0.142	2.309	10.106
mg MDA/kg	SNV-DE SG 1,4,4,1	8	124	0.746	0.213	0.342	0.338	1.247	4.376
nmol Carbonyls/mg protein	SNV-DE SG 1,4,4,1	4	126	0.441	0.343	0.065	0.499	1.046	4.832

^1^ Fatty acids were expressed as g/100 g fatty acid methyl esters. DM = dry matter; MDA = malondialdehyde; C16:0 = palmitic acid; C18:0 = stearic acid; C18:1 n-9 = oleic acid; C18:2 n-6 = linoleic acid; C18:3 n-3 = linolenic acid; SNV = standard normal variate; DE = de-trending; SG = Savitzky–Golay derivates; LVs = latent variables; *n* = number of samples; 1VR = coefficient of determination in cross-validation; RMSECV = root mean square error of cross validation; RPD_CV_ = residual prediction deviation in cross validation; RER_CV_ = range error ratio in cross validation; R^2^v = determination coefficient of external validation; RMSEV = root mean square error of validation; RPDv = residual prediction deviation in external validation; RERv = range error ratio in external validation.

## Data Availability

Data sharing is not applicable.

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
