# Peer review of "Near-Infrared Spectroscopy (NIRS) as a Tool for Classification of Pre-Sliced Iberian Salchichón, Modified Atmosphere Packaged (MAP) According to the Official Commercial Categories of Raw Meat"

_foods, 2021, doi:10.3390/foods10081865_

Round 1

Reviewer 1 Report

The manuscript entitled “Near Infrared Spectroscopy (NIRS) as a tool for classification pre-sliced Iberian salsichón modified atmosphere packaged (MAP) according to the official commercial categories of raw meat” describes an interesting approach for the classification of these samples according to its commercial category and also for the quantification of some parameters. For classification purposes, three different chemometric tools (PLS-DA, SIMCA, LDA) were tested while for quantification purposes only partial least squares was used. The amount of analysed samples is enough for obtaining reliable conclusions. The article’s writing is good, although some minor mistakes were detected.

I have some doubts/suggestions that I think they should be properly addressed to increase the quality of this work.

-The developed PLS-DA was based on the PLS-1 or PLS-2 algorithm?

-The analysis of the regression coefficient vectors will be interesting to understand the best spectral regions for the developed PLS-DA and PLS models.

- In section 3.2 regarding PLS-DA, the authors refer that black category yielded a low SP in the external validation…Was it misclassified as red or white category? This information is interesting….

-In section 3.2 regarding SIMCA, the authors refer that the SP values were low as expected (line 317)…Why? Please add this to the text.

- Do the authors have any explanation why the obtained results for LDA were much better than the ones obtained with PLS-DA and SIMCA?

-The values of the external set should be within the values of the calibration set…In this sense, the sample division should be rectified as for alpha and C18:3 n-3 the external set has values higher than the calibration set…

-in the discussion section (line 406), how the authors can refer good accuracy of the PLS-DA models when the SE values in the validation set were below 50 for all categories and the best SP value in the validation set was only 78 for the white category? In my opinion the results obtained through PLS-DA were not good…

-regarding the quantification models, do the authors have any explanation why the results obtained for the validation set are much worse than the ones obtained for the calibration set?

-the RPD and RER parameters by definition should only be calculated with the validation set results…please remove these results for the calibration set because it is not correct

-in relation to the quantitative predictions only the parameter C18:3n-3 yielded a R2V higher than 0.8 and a RPD and RER higher than 2 and 10, respectively. This means that it was the only model with a good predictive ability…therefore the discussion should be changed…

-a figure with a sample, NIR instrument and with the respective NIR spectra will be interesting for the readers to visualize the developed work.

- testing different pre-processing techniques are important to improve the results of the models but testing different spectral regions could also improve the developed models...why the authors did not tested different spectral regions?

About the writing I have some suggestions:

-introduction section

                - line 27: replace “which enjoy of great” with “which have a great”

                -line 56: replace “been yet addressed.” with “ been addressed yet.”

                -line 59: replace “are more in line with” with “are more convenient with”

                -line 60: replace “because of the product” with “because the product”

                -line 65: add an end stop to the sentence.

-materials and methods section

                -line 110: the Celsius degrees unit symbol is not correct

                -line 247: replace “performance of models” with “performance of the models”

                -line 250: replace “was calculated” with “was reflected”

                -lines 268, 271, 288: remove the space between paragraphs

-results section

                -line 297 and 298: rephrase this sentence.

                -line 315: replace “the most suitable discriminant equation” with “the best discriminant model”

                -line 328: add an end stop to the sentence.

                -lines 361: replace “the exception of models for” with “ the exception of C18:1 n-9 and linoleic acid (c18:2 n-6) models”

-discussion section

                -line 388: please put less label values in the x-axis of figure S2.

                -line 403: replace “those in the areas above-mentioned” with “those in the abovementioned areas”.

                -line 412: replace “as they achieve the highest” with “as they have the highest”

                -line 441: replace “ In refer to” with “In relation to”

                -line 463: replace “So, lower would” with “ So, worst results would”

-conclusion section

                -line 483: replace “useful as control tool” with “useful as a control tool”

Author Response

The author´s replay to Reviewer may is presented in the attached file

Reviewer 2 Report

The authors present a study on the usability of NIR spectra obtained thru the pack of Spanish fermented sausages to i) classify the production system of the raw material (Iberian Quality Standard, 3 levels) and ii) to quantify certain quality parameters such as dry matter, fat and lipid oxidation markers and fatty acid composition. Therefore, NIR spectra were obtained of several packs per quality class over the course of 12 months of cool storage.

As such, the subject appears to be relevant to the readership of FOODS as it showcases the use non-destructive testing for monitoring quality changes during storage and for preventing food fraud (fraudulent labelling). The use of NIR is not entirely novel, while reports on its application through the pack of meat/meat products is scarce.

Overall the experiment is well described. Chemometric procedures are well explained (minor exceptions see below) such as readers not that familiar with it can follow. An independent data set was used to validate the prediction performance.

Yet I don’t see the point in presenting a great variety of chemometric approaches that yield results of little to no practical use. Instead I suggest to focus more on the most promising qualitative and quantitative model(s) and explain in greater detail why these models work. For example, the loadings (ie. spectral contributions) could be shown and discussed. Also the robustness of the models should be assessed and discussed in greater detail instead. When looking at the validation parameters it appears from the R²drop (likewise, drop in RPD; SE; SP) that calibration models seem to be over fit. For example, to predict DM, the best model needs 10 latent variables (LV) which makes me doubt that it can be used in practice. Instead, a 3 LV model based on (SG 1,4,4,1) appears much more robust with only slight loss in prediction performance. A thorough investigation of the loadings could explain why this is the case. This said, it would be useful to present descriptive statistics of the reference data (DM, PUFA etc.) for the three studies quality classes (see comment below). Plus, I suggest adding a PCA analysis of the raw and or pre-treated spectra with the storage duration and quality class a passive variables such as one can see i) if/how quality classes or storage time group based on the spectra and b) which spectral components contribute to these groupings. I suggest the conclusions to be revised accordingly.

This said, I am wondering whether the global approach (relatively few samples WITHIN three quality classes sampled over several time points) is suitable. Wouldn’t it be more successful to model WITHIN a given time point? Else changes that occur over time could overlap changes between commercial categories.  

Careful edition of the use of English (mainly grammar) is suggested.

Minor/detailed comments:

Line 87 ff. it remains unclear whether “several packages” refer to several batches (i.e. from different production dates, LOT codes or alike). Preferably a large variability is included in such calibration/validation trials. Thus, authors are asked to please elaborate on the background of the samples.

101 ff: do I understand correctly that the different categories (black, red etc.) were each dried for a different period, i.e. black is dried much longer than white? If so, the a clear hypothesis could be derived (NIRS could well be able to predict the varying water content).

L 139: please explain what “manual and random selection” means. On what ground were the samples selected?

143: should it probably read “during the performance evaluation”? (insert)

188: shouldn’t this be R² ? (else explain the abbreviation “1-VR”)

206: table 2 should be placed in the results section

246 shouldn’t this read “carbonyls”?

282: table 6 doesn’t show the descriptive statistics of the reference parameters but results of NIR models. Should read table 5 instead. Here it would be interesting/relevant to see the results split be black/red/white label products.

297f sentence doesn’t make (full) sense to me. Please rephrase

304 derived from

313: I find it irrelevant to show the SIMCA results as the have little practical use. I suggest to describe only verbally that SIMCA was also used but yielded results of little value.

434: I miss a discussion (and clear conclusion) which classification approach is deemed suitable

Note: Spectra data/information should read spectral information/data

Author Response

(The authors gave the same response as above.)
